# R–R–T (resistance–resilience–transformation) typology reveals differential conservation approaches across ecosystems and time

Guillaume Peterson St-Laurent [1,2 ✉], Lauren E. Oakes [2,3,4 ✉], Molly Cross[2,3] & Shannon Hagerman[1,2]

Conservation practices during the first decade of the millennium predominantly focused on resisting changes and maintaining historical or current conditions, but ever-increasing impacts from climate change have highlighted the need for transformative action. However, little empirical evidence exists on what kinds of conservation actions aimed specifically at climate change adaptation are being implemented in practice, let alone how transformative these actions are. In response, we propose and trial a novel typology—the R–R–T scale, which improves on existing concepts of Resistance, Resilience, and Transformation—that enables the practical application of contested terms and the empirical assessment of whether and to what extent a shift toward transformative action is occurring. When applying the R–R–T scale to a case study of 104 adaptation projects funded since 2011, we find a trend towards transformation that varies across ecosystems. Our results reveal that perceptions about the acceptance of novel interventions in principle are beginning to be expressed in practice.

[1] Faculty of Forestry, University of British Columbia, 2900 – 2424 Main Mall, Vancouver, BC V6T 1Z4, Canada. [2] Climate Change Specialist Group, Species Survival Commission, International Union for Conservation of Nature, Rue Mauverney 28, 1196 Gland, Switzerland. [3] Wildlife Conservation Society, 1050 East Main Street, Suite 2, Bozeman, MT 59715, USA. [4] Department of Earth System Science, Stanford University, 473 Via Ortega, Stanford, CA 59715, USA. ✉email: Peterson.guil@gmail.com; loakes@wcs.org

Unprecedented changes in climate[1] and biodiversity loss[2,3] have increased recognition for the need and urgency to manage climate risks across all spheres of society. In 2019, the Global Commission on Adaptation released a report highlighting the human, environmental and economic "imperatives for accelerating adaptation"[4]. This heightened recognition about the need for adaptation is further reflected in global efforts to conserve biodiversity, including initiatives led by the International Union for Conservation of Nature[5], for example, and various decisions, programs, and thematic areas under the Convention on Biological Diversity. The Intergovernmental Science-Policy Platform on Biodiversity and Ecosystem Services global assessment also highlights the need for climate adaptation to support future pathways that improve outcomes for both biodiversity and societal objectives[6].

Within these and other conservation institutions that operate at multiple scales, novel conservation actions and objectives aimed specifically at helping ecosystems adapt to the accumulating impacts of climate change (hereafter "conservation adaptation") have gained increasing attention[7–9]. For example, transformative actions such as species translocation and objective setting by triage principles that were not long ago eschewed by most conservation scientists, and declared by practitioners and policy-makers as anathema to the practice of conservation[10,11], are today increasingly highlighted as necessary components of conservation adaptation[12–16]. At the core of arguments for a more future-looking, transformative approach are concerns that contemporary conservation practices—even those informed by climate change—remain focused on actions and objectives to preserve historic conditions rather than facilitating transitions to anticipated new ecological and climatic regimes[17–20].

Appeals for a "shift toward managing change rather than building resilience"[21] are gaining traction in the academy, but concerns remain that the changes required to conserve biodiversity given the challenges of climate change have not yet been integrated into practice[7,22]. Previous research suggests that this hesitancy—observed for both experts and publics alike—is a function of the ecological, values-based, and institutional complexities that are inextricably bound in conservation practice[8,23]. However, few empirical attempts have systematically categorized on-the-ground conservation adaptation projects to assess the extent to which adaptation actions are being implemented, by what approaches, and where they are occurring[24]. None that we are aware of have assessed how transformative (or not) these actions are in terms of adopting a future-looking approach instead of maintaining current or historical conditions. Previous work has identified classification frameworks and typologies to synthesize and organize conservation adaptation actions based on types of actions and/or desired outcomes. In addition to providing a much-needed structure and guidance to help practitioners and managers navigate the "sea of adaptation ideas and recommendations"[25], these frameworks can also help assess trends in adaptation actions, categorize decisions, and evaluate trade-offs (e.g., adopting future-looking approaches versus maintaining current or historical conditions, species versus landscape level conservation). However, existing typologies tend to be broad, ambiguous in terminology that are open to multiple potential interpretations, and difficult to operationalize.

This study takes a critical first step towards answering the question of whether, and to what extent, a shift toward transformative actions is occurring within the field of conservation adaptation. We make two novel contributions. First, we develop a typology of adaptation actions that reduces linguistic uncertainty[26] and supports subsequent, widespread empirical analysis of adaptation trends in the field of conservation practice. Second, we trial the typology by applying it to a case study of 104 adaptation projects funded by the Wildlife Conservation Society (WCS) Climate Adaptation Fund (hereafter "CAF projects") in the United States to assess potentially emerging trends in the field of conservation adaptation between 2011 and 2019. This study addresses two questions in relation to the CAF projects dataset: (1) What types of adaptation actions have been funded and implemented between 2011–2019, and (2) To what extent have adaptation actions changed over time, and do they vary across ecosystems in which they are implemented?

CAF projects, supported in-part by $19 million in funds (to date) from the Doris Duke Charitable Foundation, have confronted diverse impacts from climate change, such as coastal erosion, drought, wildfire, and flooding, in a range of terrestrial, coastal and aquatic ecosystems across 40 states and territories. Funding decisions are made through a systematic, independent review process that involves scientists and staff at WCS and members of an Advisory Board comprised of representative experts from the adaptation field. The field of conservation practice is diverse and with varied contours as it is pursued by different organizations, governments, including Indigenous governments, in different regions and contexts globally. While the portfolio of CAF projects does not represent the full range of this diversity, it provides a singular dataset of conservation projects that focuses specifically on adaptation objectives and actions across time and as pursued within diverse ecosystems. As such, the portfolio of CAF projects offers a unique "learning laboratory" of adaptation efforts from early adopters and represents an ideal case study to test our typology.

## Typologies of adaptation

Land managers, conservation practitioners, and researchers face a multitude of options when deciding how to tackle climate change at different scales[14,27,28]. Combined with the complex nature of adaptation[29], the multiplicity of adaptation actions can be a double-edged sword, making it challenging for decision-makers and managers to identify and consider the trade-offs between available options[25]. Furthermore, efforts to communicate and learn about different adaptation approaches is complicated by the varied uses and interpretations of commonly applied terms. The term resilience is one relevant example of this linguistic imprecision. Resilience, which "has begun to rival 'sustainability' as an environmental buzzword"[22], is "being used—and overused"[7] to the point where "it is in danger of losing clear meaning"[30]. Its meaning still remains unclear in many contexts; thus, recent attempts have been made to define what resilience is and is not[31,32]. Others, such as Fisichelli et al.[33], urge scholars and practitioners to move beyond its usage, arguing that the term has become increasingly vague with meanings ranging across a spectrum from resisting changes, absorbing changes, and even allowing for transformative changes through self-organization.

A precise definition of terms is critical to developing typologies (i.e., classifications based on types of interventions and/or conservation objectives) in rapidly evolving fields of practices, such as contemporary conservation adaptation. These typologies then represent powerful tools that enable comparisons between approaches and outcomes. They can also generate analytical insights by allowing researchers and practitioners to conceptualize, measure, and synthesize changes and/or differences over time, and between geographies or types of ecosystems. Perhaps the most commonly used approach to categorize adaptation actions is the grouping of overarching categories of actions (e.g., restore degraded or create new protected areas, enhance connectivity, protect climate refugia)[27,34,35]. This approach is appealing and straightforward given its simplicity. However, other typologies such as those that organize adaptation actions

based on conceptual hierarchy can provide greater analytical potential and reproducibility.

The most commonly applied and, arguably, useful typologies based on conceptual hierarchies have categorized conservation adaptation actions on a change continuum ranging from resistance to transformation. This continuum typically involves three categories: (1) resisting changes in order to maintain current conditions (Resistance); (2) improving the capacity of a system to return to desired conditions after disturbance (Resilience); (3) allowing and/or facilitating the transition to new conditions (Transformation)[18,33,36–38]. Other similar terminologies also exist —for instance, North American researchers and agencies have proposed the Resist, Accept, or Direct (RAD) framework[39]. The main difference between the two typologies is that the RAD framework emphasizes on managers' actions, whereas the Resistance-Resilience-Transformation framework proposes a combination of actions and outcomes. Also, there is a difference between the intermediate category: Resilience focuses on enhancing the capacity of a system to return to a desired state, whereas Accept (RAD framework) refers to allowing changes to occur without interference. The RAD framework glosses over the less-transformative end of the spectrum by leaving resilience out entirely and not differentiating between resistance and resilience.

Researchers have also synthesized adaptation actions based on their level of departure from business as usual[7], how they reflect the adaptive capacity of a species or population[40], or how they embrace change and novelty[31]. For example, Watson et al.[41] identify the following: (1) continuing 'best practice', (2) extending on 'best practice' with information about species and ecosystem responses to past climate change, and (3) integrating climate information into future planning. Similarly, Cross et al.[42] have proposed a framework focused on climate information—the "what, when, where, and why of climate-informed action"—to appraise departure from business-as-usual. Others have surveyed adaptation actions depending on the level of risks and uncertainties associated with their implementation (e.g., from risk-adverse to risk-tolerant)[25]. More recently, Prober et al., (2019)[43] proposed a typology for adaptation actions in terrestrial ecosystems that is organized in a matrix with two axes forming four quadrants. The first axis draws on the core ecological mechanisms of the adaptation actions, whereas the second axis considers the level of intrusiveness and degree to which the action is 'climate-targeted' (i.e., done notably differently than actions designed and implemented under an assumption of static climate conditions).

While conceptually useful, the typologies presented above often include concepts that are used interchangeably or are loosely defined (exceptions include[39,40,43]). This linguistic imprecision adds another layer of uncertainty to an already-complex field of practice. It limits the analytical potential, real-world applicability and replicability of the typologies[26,44], ultimately preventing cross-comparison between adaptation projects, studies, regions, and ecosystems. Furthermore, many existing reviews of adaptation actions have limited scope, for instance, by focusing only on forests[19,36] or terrestrial systems[43].

**The resistance–resilience–transformation (R–R–T) Scale**
We propose a novel application of the concepts of resistance, resilience, and transformation (hereafter referred to as the "R–R–T scale"; Fig. 1; Table 1). To solve issues of linguistic uncertainty, our scale enables sharper resolution between well-defined and delimited concepts that is applicable to a broad range of ecosystems. We constructed the typology as a continuous interval scale—as opposed to one using nominal indicators without clear hierarchical relations. Ranging from active resistance to accelerated transformation, our six-point scale is

designed on a continuum representing progressively greater acceptance of changes in ecosystem structure and function. It enables a finer-scale assessment of trends in the degree to which adaptation actions in the conservation community may be shifting toward transformative action, as well as a practical application of contested terms, such as resilience. This finer resolution of the R–R–T scale—six categories instead of three as in the original framework—facilitates the classification of objectives on the spectrum from resistance to transformation.

Recent critics have pointed out that the concepts of resistance, resilience, and transformation do not represent fundamental alternatives because they can materialize simultaneously from the same intervention, but at difference scales (e.g., actions that generate resistance at one scale can lead to resilience outcomes at another scale)[43,45]. In response, our typology considers primarily the objectives of the adaptation project. Categorizing on-the-ground outcomes after project implementation, in contrast, can prove challenging[43], as certain actions can also lead to unanticipated outcomes across scales (more details in discussion section).

We further distinguished the concepts of resistance and transformation into two and three categories, respectively, to refine their scope and capture distinct attributes of adaptation actions. This decision also allowed for defining resilience more precisely, to address its broad and inconsistent used in theory and practice. The first two levels of the R–R–T scale, active and passive resistance, refer to interventions aimed at actively (i.e., through direct and proactive management) or passively (i.e., through indirect interventions with no active management) resist the changes brought by climate change. The third level, resilience, describes interventions that enhance the capacity of ecosystems to return to desired conditions (past or present) after a disturbance. This definition, which aligns closely with the original ecological meaning of the term[30], implies that resilience shares the end goal with resistance of generally limiting changes, but it acknowledges that some changes are unavoidable and sometimes desirable. For instance, restoring forest ecosystems with a diversity of native species can increase resilience (i.e., because there is a greater likelihood that some tree species will continue to persist and function during a disturbance like drought or fire), but it may also lead to changes in community composition. Similarly, the introduction of ecosystem engineers such as beaver can increase the resilience of an ecosystem (i.e., the reconnected floodplain can absorb more water during a flood and reduce flood-related damage to downstream reaches) while also altering historical conditions of streams and valley bottoms[46,47]. A survey of published empirical studies[48] suggests that resilience is commonly used to represent resistance, or recovery, or both. Our application of the term emphasizes recovery, while acknowledging the inevitability of some new elements.

The last three levels represent different degrees of transformation. Autonomous transformation describes actions aimed at allowing for changes without actively shaping the projected transformation[33], which is equivalent to the RAD framework's Accept category[39]. Directed and accelerated transformation aim to drive a shift towards future projected conditions; they are distinguished by the relative speed at which transformation occurs. Directed transformation encompasses actions delivering the first few steps of the anticipated transformation, such as translocating species into areas that are expected to be suitable in the future and are also located within their current native range, but outside of their genetic range (e.g., assisted gene flow)[49,50]. In contrast, accelerated transformation denotes a jumpstart to the anticipated conditions, such as translocating species to areas that are anticipated to be climatically suitable in the distant future and that are located outside of their current native range (e.g., assisted range expansion or assisted colonization)[15,51,52].

## TRANSFORMATION

**6** **ACCELERATED TRANSFORMATION**
Actions designed to more rapidly advance transition towards new structures and functions.

**5** **DIRECTED TRANSFORMATION**
Actions designed to drive transition towards new structures and functions.

**4** **AUTONOMOUS TRANSFORMATION**
Actions designed to facilitate the autonomous transition to new structures and functions.

**3** **RESILIENCE**
Actions designed to improve the capacity of a system to return to desired past or current structures and functions following a disturbance to the extent possible while recognizing some new elements are inevitable.

**2** **PASSIVE RESISTANCE**
Actions designed to passively maintain current/ historical structures and functions.

**1** **ACTIVE RESISTANCE**
Actions designed to actively maintain current/ historical structures and functions.

## RESISTANCE

**Fig. 1 Resistance–resilience–transformation (R–R–T) scale with definitions.** The R–R–T scale is a six-point continuous interval scale representing a continuum spanning from actively resisting changes to accelerating transformation towards new, more climate-adapted conditions.

### Results: characterizing changes in practice

Overall, we find that CAF projects funded prior to 2016 focused primarily on the resistance–resilience end of the R–R–T scale, and those implemented afterwards were more likely to involve transformation. The types of approaches differ across ecosystems with more resistance projects occurring in deserts, grasslands and savannahs, and inland aquatic ecosystems, and more transformative projects in forest, coastal aquatic, and urban/suburban ecosystems.

Supplementary Data 1 includes the R–R–T scores for the 104 CAF projects, along with short project descriptions that were crafted, on an annual basis, from full proposals by WCS staff to provide a brief abstract of the projects for the general public. Of the 104 projects funded between 2011 and 2019, the most common categories of actions were resilience (40%), autonomous (26%) and directed (18%) transformation, and active resistance (10%, Fig. 2). The significant difference (Kruskal–Wallis: H(8) = 12.5, $p < 0.01$) found between the mean R–R–T score of projects funded in 2017–2019 (M = 4.08, SD = 1.15) compared to 2011–2014 (M =

2.93, SD = 1.21) and 2014-2016 (M = 3.42, SD = 1.06), combined with the upward trend observed in yearly mean R–R–T scores (Fig. 3), suggest a shift towards transformation over time. Fisher's exact test revealed that the percentage of transformative projects differed by years ($p = 0.01$; Fig. 2), with the period 2011–2013 (40%) and 2014–2016 (39%) having proportionally less than in 2017–2019 (64%). In contrast, most active resistance and all autonomous resistance projects were funded in early years (2011–2013), whereas the years 2014-2016 predominantly focused on resilience (53%). Figure 3 indicates a demarcation between the projects funded before 2016, which lean towards the resistance–resilience end of the R–R–T scale, and the projects funded afterwards that are more inclined towards transformation.

Adaptation approaches also differed by ecosystems (Fig. 3). We found a difference between the average R–R–T scores of projects conducted in different ecosystems (*Kruskal-Wallis*: H (6) = 12.62, $p < 0.05$), with forests receiving a higher ($p < 0.05$) average R–R–T score (M = 4.06, SD = 1.01) than inland aquatic ecosystems (M = 3.26, SD = 1.21). Figure 2 further indicates

**Table 1 Examples of adaptation actions and their primary objective for the six categories of the "R–R–T scale".**

| Categories | Examples of actions | Primary objective |
|---|---|---|
| 1. Active Resistance | Eradicate non-native species in grassland or forest ecosystems. Install and manage water control structures to maintain historic water levels in a coastal impoundment. | Actively prevent changes in species composition. Actively resist rising sea levels. |
| 2. Passive Resistance | Create or expand protected areas in climate refugia. Purchase conservation easements to protect a species that is endangered by climate change. | Passively maintain current ecosystems. Passively protect species in their historical habitat. |
| 3. Resilience | Reconnect previously existing corridors to allow the migration of specific species. Restore streams by re-introducing beaver. | Enhance the ability of species to persist as climate changes by removing barriers to movement and dispersal Increase resilience of stream functions to natural disturbances such as floods and droughts. |
| 4. Autonomous Transformation | Connect relatively warmer and colder aquatic areas. Apply forestry techniques designed to increase native species diversity. | Create opportunities for species movements to seek cold water refugia. Increase chances that some species will thrive as climate changes. |
| 5. Directed Transformation | Use assisted migration by planting with seeds gathered in a warmer part of a species' current range (aka assisted gene flow). Use climate-informed forestry to direct future species composition (e.g., post-harvest planting using drought-resistant native species). | Drive transition towards climate-adapted genetic composition of species or populations. Drive transition towards more climate-adapted native species compositions. |
| 6. Accelerated Transformation | Use assisted migration to move a species outside of its current or historic range (aka assisted range expansion). Restore riparian ecosystems by inoculating soils with non-native inoculant materials that are adapted to warmer and dryer conditions. | Accelerate climate-driven species transition. Accelerate transition towards more climate-adapted ecosystem functions. |

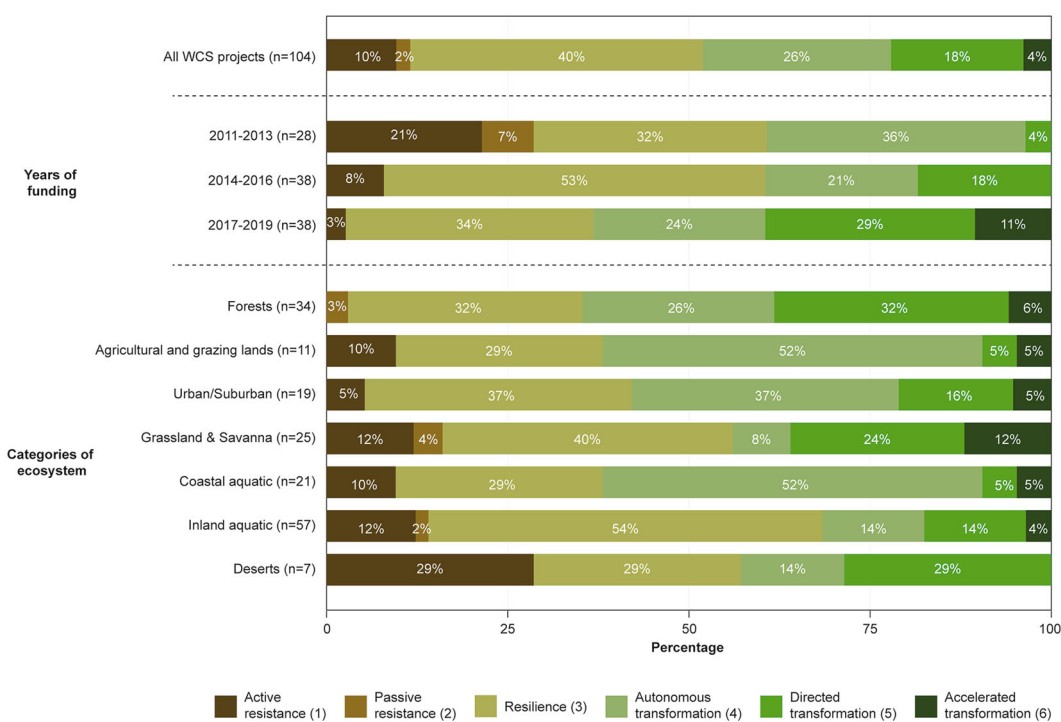

**Fig. 2 Distribution of categories of adaptation projects using the R–R–T scale by years of funding (using groupings of three years) and type of ecosystems in which the projects were conducted.** Some projects have been implemented in more than one type of ecosystem.

that, proportionally, most active resistance projects were conducted in deserts, grasslands and savannas, and inland aquatic ecosystems. In contrast, the CAF program did not fund any active resistance projects in forests. More than half of the projects conducted in inland aquatic ecosystems aimed at resilience, whereas a similar proportion of projects in coastal aquatic ecosystems and agricultural and grazing lands focused on autonomous transformation. The proportion of projects that involve some level of transformation did differ by ecosystems ($p = 0.04$, *Fisher's exact test*), with a significantly higher

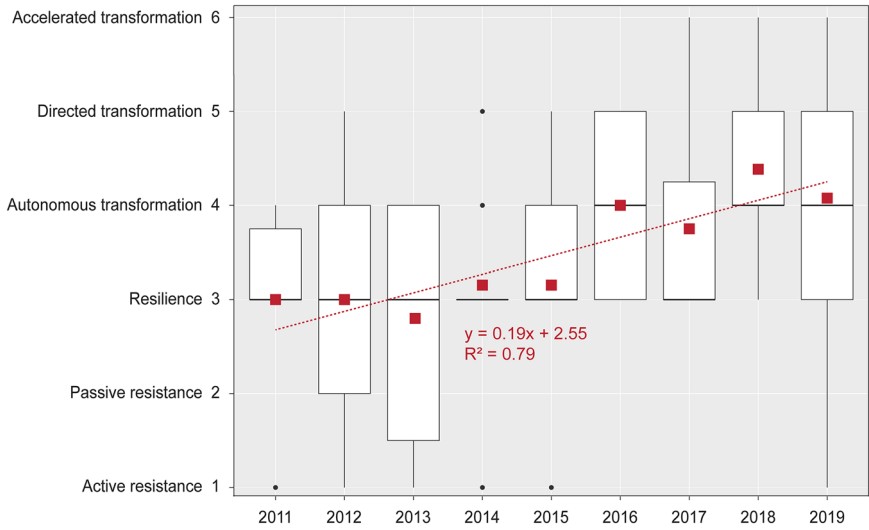

**Fig. 3 Boxplot showing the distribution of R–R–T scores of CAF projects funded between 2011 and 2019.** The center lines correspond to the median, the lower and upper hinges to the first and third quartiles, and the upper and lower whiskers to the maximum and minimum values (no further than 1.5x interquartile range). Outliers are shown with black circles, and yearly mean scores with red squares. A best-fit linear trendline with its equation and R-squared value is shown in red.

proportion in forest (64%), coastal aquatic (62%) and urban/suburban (58%) ecosystems compared to inland aquatic ecosystems (32%).

**Discussion: implications for conservation practice**

Reflecting broader conversations about the need for more transformation actions in conservation adaptation[7,8,22], earlier CAF projects (prior to 2016) were more likely to include approaches designed to resist changes, whereas more recent projects tended towards managing for transformative change. This finding suggests that what Prober et al. (2019)[43] refers to as a "renovation paradigm"—that climate informed actions share the common goal of repairing or improving upon something, or imparting new vigor—is beginning to be observed in practice.

Our results also indicate that projects conducted in certain ecosystems were more likely to involve specific categories of adaptation actions (e.g., resilience in inland aquatic ecosystems, autonomous transformation in coastal aquatic and agricultural and grazing lands). Conservation in forested systems appears to be at the leading edge for the application of transformative actions (64% of projects), particularly in the form of assisted migration within current natural range. Most directed and accelerated transformation projects in other ecosystems (e.g., inland riparian, grasslands and savanna) also involved components of assisted migration of trees or plants (exceptions include, for instance, the assisted migration of seabird species in Hawaii). This finding is reflective of previous research that identified plants as a susceptible taxon for assisted migration because it involves a lower risk of intracontinental invasion due to dispersal constraints[53]. In contrast, a smaller percentage of transformative actions were implemented in inland aquatic ecosystems. This relative conservatism for this ecosystem type may indicate that the conservation adaptation community is still defining and exploring what transformative action could be like in these systems, although actions such as the assisted migration of freshwater species are already being discussed and implemented in some regions[54]. It may also indicate that there is a less urgent need for managing for change in some ecosystems, or perhaps that the extent and availability of research to inform future-looking action differs across ecosystem types.

Although six mutually-exclusive categories comprise the linear R–R–T scale, we acknowledge that the distinction between categories (e.g., passive resistance, resilience and autonomous transformation) is not always black and white. In particular, the potential outcomes of adaptation actions may overlap two or more categories. For instance, a project that enhances longitudinal connectivity of forested or aquatic systems may ultimately result in re-establishing historical connectivity[55] (active resistance), enhancing the capacity of a system to respond to disturbance (resilience) and/or facilitating the migration of species to new areas[56] (passive transformation). Similarly, the reintroduction of ecosystem engineers (e.g., beavers) may allow a system to better resist changes, but it may also enhance its resilience by promoting recovery after natural disturbances[47,57]. To prevent this ambiguity, we classified projects based on the primary objective identified in the proposals with regards to desired future conditions (and how similar or different they are from current or historic conditions). The R–R–T scale could also be used to assess the on-the-ground outcomes of adaptation actions after their implementation, which could lead to situations where one project could be classified into multiple categories at once.

Our focus on the portfolio of 104 CAF projects offered a data-rich, unique opportunity to trial a novel framework for conservation adaptation and provide empirical evidence suggestive of trends in conservation practice that have been anecdotally observed globally and across diverse groups[7–9,15,22,43]. At the same time, these results, building as they do on the activities within one funding portfolio in the United States, come with some important caveats. Foremost is caution in generalizing the findings from this case to the broader field of conservation (something that we do not claim here). To our knowledge, the CAF portfolio represents one of the largest pools of conservation projects that are specifically and intentionally designed to achieve adaptation outcomes. This portfolio is embedded within, but is not considered a random subsample of conservation adaptation projects conducted in the United States, nor is it representative of the field of practice. The unavailability of other similar datasets—which speaks directly to the necessity for more empirical assessments of conservation adaptation actions—limited our analysis to the use of the CAF projects as proxy to assess trends in conservation adaptation funding and implementation over the last

decade. Further research is needed to apply the R–R–T scale to explore its application to other adaptation contexts; its application to a global fund could be of particular interest. It could also be used to assess conventional conservation projects with no specific focus on adaptation to elucidate trends in the broader field of conservation, if such a dataset could be assembled.

In addition, the evolution of funding guidelines for the Fund also affected the results of this study. The decision to stop funding land protection projects (e.g., purchasing land or easements) and focus solely on active management and restoration projects after 2012 could help explain the low overall number of passive resistance actions. Such projects have still taken place in the field of practice—at large—but were no longer captured in the sample. Similarly, in its early years the Fund prohibited any project proposal that involved the movement of species outside of their current natural range. The slow occurrence of accelerated transformation projects since 2016—the year when assisted range expansion was deemed acceptable by the Fund—suggests a potential upsurge of such projects in the coming years. These funding rules offer additional evidence of the shift in perspective from resistance to transformation and the more widespread acceptance of controversial transformative projects (e.g., assisted migration).

An increasing number of scholars advocate for a better understanding of the human and social components of adaptation, and most notably the worldviews and perceptions of the different actors who are involved in, or could be affected by these processes[58–61]. Recent studies suggest that the general public[62] and forest practitioners[63] in Canada, for example, generally support the concept of assisted migration in the forests, with a preference for its implementation within natural range compared to outside of natural range. While these results reveal enduring worries about the transgression of natural boundaries[23], it also provides evidence of shifting management practices, with accelerated transformation possibly representing the next frontier of conservation. Similarly, findings from our study also suggest that perceptions about the acceptance of novel interventions in principle[28,62,63] are beginning to be expressed in practice. While not representative of widespread practice within the conservation community, our results reveal a possible shift in perceptions of early adopters—both the adaptation experts advising and running the WCS Climate Adaptation Fund and the on-the-ground practitioners involved in CAF projects—towards a greater willingness to embrace the concept of transformative change and support projects in this category of action.

There are no one-size-fits-all solutions to climate change adaptation for conservation. Precautionary, "low risk" actions aimed at resistance and resilience—such as protecting intact ecosystems[64,65]—are valuable, particularly when managers work under high uncertainties and have access to little information about future climate projections and the associated local impacts[25]. Yet, degraded ecosystems or working landscapes may require more transformative actions and the public support to do so, in effort to meet the shifting goals in a changing climate. Our study provides empirical evidence of paradigm shift, as practitioners and funders begin to move in this direction.

## Methods

We carried out a content analysis[66] of full proposals for projects that were awarded grants from the Climate Adaptation Fund (CAF) to categorize the portfolio of CAF projects using the qualitative analysis software NVIVO (version 12.6.0) and coding project documents directly using the R–R–T scale. To ensure consistency, reduce ambiguity and capture the motivation behind adaptation actions, each project was assigned one of the six categories of the R–R–T scale based on their primary objective.

Before starting this content analysis of proposals and after several iterations, we developed the first version of the R–R–T scale in the form of a 5-point continuous interval scale (i.e., initially without the "accelerated transformation" category). We performed three rounds of coding, each time using an updated version of the R–R–T scale. We separated the first round of coding into four phases that facilitated a staggered approach of independent coding, group discussion, and iteration of the definitions through analysis of subsets of the full portfolio. During these phases, three team members (including the first two authors) independently coded the full proposals of CAF projects funded during groupings of two or three years (e.g., 2011 and 2019). Three projects that were initially funded did not complete implementation, but we still included them in our analysis. Across all projects, we coded for funded activities; if any activities were removed between the proposal and granting stage, we did not include those activities in our coding of the project. After each phase, the coders compared their results, discussed any disagreements, and reached consensus on a final score for each project. Throughout the process, the team members revised the R–R–T scale (e.g., refining definitions, examples) and the scores from previous phases.

During the second round of coding, a fourth team member (third author) coded all the CAF projects using the latest iteration of the 5-point R–R–T scale. After cross-referencing and identifying divergence in the scores from rounds one and two, we deliberated and reached consensus on a score for each project. As a result of this coding round, the full research team reviewed the R–R–T scale and added a sixth category to account for an emergent level of transformation, making it a 6-point continuous interval scale. In the third and final round of coding, the two first authors independently coded the projects again with the revised scale, reaching intercoder reliability of 95%. We discussed and agreed upon a score for any projects where discrepancies remained.

**Statistics and reproducibility**. Statistical analyses were conducted in R studio[67] (Version 1.2.1335). We used descriptive statistics like means and proportions to summarize the dataset of conservation adaptation projects ($n = 104$). We used a Kruskal-Wallis test with Dunn's post hoc test controlled with Bonferroni adjustment to compare average R–R–T scores by years and ecosystems in which the projects were implemented. We also used Fisher's exact test with post hoc test adjusted by FDR method for multiple comparisons (Benjamini–Hochberg false discovery rate) to compare proportion of projects that involve some level of transformation by years and ecosystems. Because of the relatively small number of projects funded each year (Supplementary Table 1), we compared the averaged R–R–T scores of the projects funded during groupings of three years, starting with 2011–2013. The portfolio of CAF projects involves a rich diversity of targeted ecosystems, intentionally sought after by the experts running the Fund. We included seven types of ecosystems in our analysis (Supplementary Table 1). Some projects were conducted in more than one ecosystem.

**Reporting summary**. Further information on research design is available in the Nature Research Reporting Summary linked to this article.

## Data availability

We have conducted our study following the BREB guidelines (Behavioral Research Ethics Board) at the University of British Columbia, Canada (Ethics ID number H19-02949). The dataset used to conduct this study does not include data actively collected by researchers with study participants. The data consists of successful grant proposals submitted by unaffiliated organizations (hereafter "grant partners") to the Wildlife Conservation Society Climate Adaptation Fund between 2011 and 2019. These proposals are owned by the Wildlife Conservation Society, but they contain grant partners' confidential data, and in some cases, intellectual property (e.g., novel technology). These proposals are therefore treated in the same way than other sensitive data collected with human subject. It is BREB's position that a breach of confidentiality of study participants (i.e., grant partners) has taken place when there is a failure to conform to the commitment that the researchers have made to the study participants when some or all the data has entered the public domain (i.e., the data has become available to any person who is not authorized to view or access the data). Thus, we shall not publicly disclose the raw research data in its original form (e.g., full grant proposals).

Supplementary Data 1 provides the year of funding, grant partner organization, type(s) of ecosystems, R–R–T score, title and short description for each of the 104 projects included in the analysis. These descriptions were crafted from full proposals by WCS staff —on an annual basis with a consistent approach—to provide a brief abstract of the projects for the general public; to provide this level of detail for each project, we include this publicly-available description. They are not always completely representative of the full scope of the funded projects, and may thus not always allow replicability of our study. However, if readers are interested in requesting the raw dataset from this publication, they can contact the lead authors, and data can be shared (removing such confidential information such as organization's financial status) upon request. They must sign a non-disclosure agreement and comply with the BREB and the Wildlife Conservation Society's guidelines for further use of the data. Readers may contact loakes@wcs.org if they want to request data.

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

## Acknowledgements

This research was financially supported by the Doris Duke Charitable Foundation. The Climate Adaptation Fund is funded by a grant to the Wildlife Conservation Society from the Doris Duke Charitable Foundation. The funder was not directly involved in conceptualization, design, data collection, analysis, decision to publish, nor preparation of the manuscript. We are very grateful to Henry Locke for his valuable support during data collection and analysis. The IUCN Species Survival Commission selected this research for inclusion in its Climate Change Specialist Group.

## Author contributions

G.P.S.L., L.E.O., M.C. and S.H.—conceptualized and designed the study. G.P.S.L., L.E.O. and M.C. collected and analyzed the data. G.P.S.L. led the writing with L.E.O., M.C. and S.H. contributing to the framing. All authors reviewed and revised the paper.

## Competing interests

The authors declare no competing interests.
