## [Peer Review File · Communications Biology]

Reviewers' comments:

Reviewer #1 (Remarks to the Author):

This manuscript covers an interesting topic about the incorporation of transformative adaptation in conservation projects. The study first proposes a novel approach to classify conservation actions according to a continuum between Resistance, Resilience and Transformation. They also use funded projects from the Wildlife Conservation Society to explore the state of the art of transformative actions occurring within the field of conservation. This manuscript has the potential to contribute to the field of conservation by providing a new, more realistic view of the Resistance, Resilience and Transformation framework. The approach and definitions are interesting, adding some degree of realism to a current existing approach. However, the analyses on the projects funded by the Wildlife Conservation Society provides a very narrow view of conservation, and therefore limiting the general scope of the manuscript. A more extensive review of other funding agencies and other local versus global conservation projects would be much more relevant.

GENERAL COMMENTS

This study would gain value if in the introduction the authors discuss the needs to incorporate adaptation into management. A more detailed explanation of how this has been integrated into different conservation acts like the Sustainable Development Goals or the Aichi Biodiversity Targets would provide a broader picture of the implications of this manuscript.

This study should be cautious about the generalisation of the results. The authors only use one funding body to estimate a tendency of the whole field of conservation. Conservation projects span from very local initiatives to more global ones. It seems unrealistic to use only one funding body from the United States, over less than ten years to explore a general tendency. Such limitation is on top of the multiple ones acknowledged by the authors in lines 220-238. Considering US actions to be representative of the field of conservation could be viewed as a lack of awareness of the diversity of conservation challenges experienced worldwide. For that reason, it would be much more relevant to compare different funding bodies to have a more representative view of the field of conservation. The authors appropriately claim for the ambiguity of the term resilience, but the definitions and examples that they provide could be improved to avoid ambiguity. It seems that from the definitions the authors provide in Figure 1, resistance (active and passive) measures are designated to maintain current or historical status, while resilience actions aim for future recovery. However, it remains unclear the separation of resistance and resilience in point three, given the interdependency of resistance and resilience. For example, actions aimed to re-introduce ecosystems engineers are going to improve the resistance to changes, as well as promote its recovery. Please check Hodgson et al. 2015 TREE, Donohue et al. 2016 EcoLetts, Ingrisich and Bahn 2018 TREE, for more extensive explanations of the interdependency between resistance and recovery. This manuscript would benefit from a clearer distinction between resistance and resilience.

SPECIFIC COMMENTS

Lines 6-8: This is an incorrect statement. There are plenty of manuscripts exploring the conservation practices applied and their success. One recent example would be Duarte et al. 2020 Nature, but there are many more.

Lines 17-18: Conservation is not only about climate change but global change. The authors should provide here, and throughout the manuscript, a clear statement of the conservation actions targeted in this manuscript.

Line 27: Could the authors please provide references supporting the increase of the need for

building resilience in science and conservation?

Lines 31-34: This sentence is very long and difficult to follow, please try to shorten it or split into two.

Line 39: Could the authors be more specific about which trade-offs?

Line 75: Given the general scope of the journal, it would be worth to define what typologies are. It would help unfamiliar readers to better comprehend the concept and the value of the research done here.

Lines 84-102: It would be good to have a more clear description of why the authors discuss these specific typologies.

Figure 1: The examples posed in this figure would be much more useful if explained in much more detail. Also, for some definitions only one example is provided, giving a narrower view in comparison with other definitions. It would be highly desirable to have a more balanced proportion of examples.

Line 143: It would be beneficial for the manuscript to provide examples of which changes could be desirable.

Figure 3: This figure does not provide estimates of the slope of the linear model tendency. At least the authors should provide the slope and if it significantly differs from 0.

Lines 162-164: Please indicate what the numbers in parenthesis represent. Are they mean and standard deviation or are they median and standard error.

Lines 196-197: This statement is generalising results coming from a very limited sampling size and only one funding body (see general comments).

Discussion: The discussion is very focused on the results coming from the analyses done from the projects funded by WCS. However, this manuscript would benefit from a more in-depth discussion of the novel R-R-T approach and present the results from the WCS funded projects as a case study.

Line 290: Please cite R software.

Reviewer #2 (Remarks to the Author):

This manuscript represents an important contribution to the rapidly evolving field of climate adaptation, particularly in refining a typology for categorizing adaptation approaches on a continuum from resistance (or persistence) to transformation (or change). The manuscript nicely describes the confusion surrounding the overly broad application of the term "resilience". Of particular interest, however is the development of a single continuous typology that integrates the concepts of passive and active management. I have attempted to do something similar (see Stein et al. 2019 -- DoD adaptation guide) through use of a four quadrat approach (change on one axis and passive/active management along the other), but find the typology presented to be a much more elegant solution and far more receptive to classifying adaptation efforts and charting the shifts in their emphasis over time.

Having done a good job of defining the existing confusion in adaptation typologies and the emphasis on activity rather than outcome-oriented categories, the manuscript tests its application on an empirical data set of adaptation projects. Although this data set has limitations, it is -- to my knowledge -- one of the only extensive sets of projects that are specifically and intentionally designed to achieve adaptation outcomes. For that reason, it is a reasonable proxy in showing trends in adaptation funding and implementation over the decade or so the program has been in existence. The manuscript addresses the one concern that I had in using the data set to chart the shift in the broader community away from resistance and towards more transformational actions -- that is, the

evolution over that period of the guidelines and expectations of the funding program and its advisors. That said, the manuscript recognizes that confounding effect, but the analysis confirms that there has indeed been a resulting shift toward more innovative and transformation-oriented projects, bolstering the conclusion that a shift in perception -- if not in the broader community then at least in this important part of the natural resource-oriented adaptation community -- is underway.

The one thing that I was expecting to see, but is missing, was any reference to a similar/related effort underway in the community towards development of the so-called "RAD" (Resist, Accept, Direct) framework, which significantly overlaps with RRT scale described here. It may be that there are no citable accounts yet for RAD, but at a minimum I would expect some recognition of that emerging framework and discussion of how RRT is similar to or different than RAD.

I strongly support publication and view this as an important contribution to the evolving science and practice of climate adaptation.

Reviewer #3 (Remarks to the Author):

1. This manuscript contributes evidence of evolution within the field of climate adaptation. I have a few concerns that could be addressed with revision, but generally feel the manuscript worthy of publication. I find the development of the typology to be an improvement of more coarse Resistance-Resilience-Transition comparisons that I have seen published, and will have broad utility. Further, the application of the typology to a subset of projects is informative of reasonably real trends in both funding and project development directions, even if only a segment of a much larger population of conservation projects. It is in my opinion that the manuscript be considered for publication following revision to address the following concerns.

2. My major concern is whether the trends documented in this paper are observed in the broader conservation world. As I read the last paragraph of the introduction (and being unfamiliar with the project proposal and selection process for the Wildlife Conservation Society's Climate Adaptation Fund), I wondered whether the project selection process increasingly emphasized transformative projects over time or prioritized those types of projects? After reading the discussion, I realize the role of funding limitations is an important factor. However, my instinct is that the trend is real. Is there additional evidence that the authors could point to that indicate that the trends here may very well be consistent with trends in the broader conservation realm?

3. Within the 'restoration' program I work with, I have noticed a transition in the types of projects considered for funding. More specifically, there seems to be a wider range of the types of projects proposed in more recent years than historically. This may partially be an effect of enhanced understanding of the ecosystem structure and function, demographic changes within the scientist and manager community that participate in the program, or the increased awareness of resilience and adaptation ideas. Anecdotally, I do think the application of the typology would see an increase from resistance to resilience in the ecosystem the program focuses on.

4. I do think the typology can be useful in understanding the trends and directions within conservation practices; however, I think there is still ambiguity in the application of the typology that could be clarified. For example, I work in riverine ecosystems and using the typology developed, have a difficult time selecting a single term for some of the management actions that are often considered. Enhancing longitudinal (upstream/downstream) connectivity could be considered resistance in that it is an action designed to reestablish historical connectivity, or it could be considered resilience as it improves the capacity of the system to reorganize following a disturbance,

or it could be viewed as transformative to facilitate a transition to a new structure and function through enabling species movements. Is the typology applied to the primary objective of the project even if it may ultimately have a different effect? Further clarification in the methods regarding how the typology coding was applied could be beneficial.

Reviewed Article: From resistance to transformation: assessing the transition to a new paradigm for conservation

1. This manuscript contributes evidence of evolution within the field of climate adaptation. I have a few concerns that could be addressed with revision, but generally feel the manuscript worthy of publication. I find the development of the typology to be an improvement of more coarse Resistance-Resilience-Transition comparisons that I have seen published, and will have broad utility. Further, the application of the typology to a subset of projects is informative of reasonably real trends in both funding and project development directions, even if only a segment of a much larger population of conservation projects. It is in my opinion that the manuscript be considered for publication following revision to address the following concerns.
2. My major concern is whether the trends documented in this paper are observed in the broader conservation world. As I read the last paragraph of the introduction (and being unfamiliar with the project proposal and selection process for the Wildlife Conservation Society's Climate Adaptation Fund), I wondered whether the project selection process increasingly emphasized transformative projects over time or prioritized those types of projects? After reading the discussion, I realize the role of funding limitations is an important factor. However, my instinct is that the trend is real. Is there additional evidence that the authors could point to that indicate that the trends here may very well be consistent with trends in the broader conservation realm?
3. Within the 'restoration' program I work with, I have noticed a transition in the types of projects considered for funding. More specifically, there seems to be a wider range of the types of projects proposed in more recent years than historically. This may partially be an effect of enhanced understanding of the ecosystem structure and function, demographic changes within the scientist and manager community that participate in the program, or the increased awareness of resilience and adaptation ideas. Anecdotally, I do think the application of the typology would see an increase from resistance to resilience in the ecosystem the program focuses on.
4. I do think the typology can be useful in understanding the trends and directions within conservation practices; however, I think there is still ambiguity in the application of the typology that could be clarified. For example, I work in riverine ecosystems and using the typology developed, have a difficult time selecting a single term for some of the management actions that are often considered. Enhancing longitudinal (upstream/downstream) connectivity could be considered resistance in that it is an action designed to reestablish historical connectivity, or it could be considered resilience as it improves the capacity of the system to reorganize following a disturbance, or it could be viewed as transformative to facilitate a transition to a new structure and function through enabling species movements. Is the typology applied to the primary objective of the project even if it may ultimately have a different effect? Further clarification in the methods regarding how the typology coding was applied could be beneficial.

Dear Reviewers,

Thank you for your thoughtful and constructive comments on our manuscript titled: “From resistance to transformation: assessing the transition to a new paradigm for conservation”. We have gone through your reports and have outlined below our response to each of your comments, with additions to the manuscript highlighted in grey.

Best regards.

Reviewer #1 (Remarks to the Author):

This manuscript covers an interesting topic about the incorporation of transformative adaptation in conservation projects. The study first proposes a novel approach to classify conservation actions according to a continuum between Resistance, Resilience and Transformation. They also use funded projects from the Wildlife Conservation Society to explore the state of the art of transformative actions occurring within the field of conservation. This manuscript has the potential to contribute to the field of conservation by providing a new, more realistic view of the Resistance, Resilience and Transformation framework. The approach and definitions are interesting, adding some degree of realism to a current existing approach. However, the analyses on the projects funded by the Wildlife Conservation Society provides a very narrow view of conservation, and therefore limiting the general scope of the manuscript. A more extensive review of other funding agencies and other local versus global conservation projects would be much more relevant.

Comments	Answer
1. This study would gain value if in the introduction the authors discuss the needs to incorporate adaptation into management. A more detailed explanation of how this has been integrated into different conservation acts like the Sustainable Development Goals or the Aichi Biodiversity Targets would provide a broader picture of the implications of this manuscript.	We added a paragraph in the introduction to specify the rise in importance of adaptation broadly and for conservation, providing three examples: IUCN, PEBES and the UN CBD. Lines 18-29: “Unprecedented changes in climate ¹ and biodiversity loss ^{2, 3} have increased recognition for the need and urgency to manage climate risks across all spheres of society. In 2019, the Global Commission on Adaptation released a report highlighting the human, environmental and economic “imperatives for accelerating adaptation” ⁴. This heightened recognition about the need for adaptation is further reflected in global efforts to conserve biodiversity, including initiatives led by the International Union for Conservation of Nature ⁵, for example, and various decisions, programs, and thematic areas under the Convention on Biological Diversity. The Intergovernmental Science-Policy Platform on Biodiversity and Ecosystem Services global assessment also highlights the need for climate adaptation to support future pathways that improve outcomes for both biodiversity and societal objectives ⁶.”

2. This study should be cautious about the generalisation of the results. The authors only use one funding body to estimate a tendency of the whole field of conservation. Conservation projects span from very local initiatives to more global ones. It seems unrealistic to use only one funding body from the United States, over less than ten years to explore a general tendency. Such limitation is on top of the multiple ones acknowledged by the authors in lines 220-238. Considering US actions to be representative of the field of conservation could be viewed as a lack of awareness of the diversity of conservation challenges experienced worldwide. For that reason, it would be much more relevant to compare different funding bodies to have a more representative view of the field of conservation.

This is an excellent comment, and one that we attempted to address early on in our study. Unfortunately, we were unable to find datasets that are equivalent to the WCS Climate Adaptation Fund (CAF). We are still in the process of securing other datasets across the world, but the process is long and is often limited by issues of data confidentiality. Waiting for more datasets could delay by at least one year our capacity to disseminate our study. Because of the novelty of the typology that we propose, and our conviction that it will have broad implications for the field of adaptation conservation and beyond, we decided to use only the portfolio of WCS CAF as a case study for the United States.

We acknowledge the limitations of this choice for our capacity to generalize. However, as pointed out by Reviewer 2, we strongly believe that the CAF projects represent one of the only extensive portfolios of projects across multiple ecosystems that are specifically and intentionally designed to achieve adaptation outcomes.

We made 3 major changes to clarify that we do not attempt to generalize our results to the broader field of conservation:

- 1) We specified throughout the paper that we focus on conservation adaptation specifically (a term that we define in the introduction).
- 2) We clarified in the introduction that we use the CAF projects as a case study to assess potential trends in the field of conservation adaptation. We also added an explanation as to why the use of the dataset is appropriate:

Lines 79-84: *“We make two novel contributions. First, we develop a typology of adaptation actions that reduces linguistic uncertainty²⁶ and that can support subsequent, widespread empirical analysis of adaptation trends in the field of conservation practice. Second, we trial the typology by applying it to a case study of 104 adaptation projects funded by the Wildlife Conservation Society (WCS) Climate Adaptation Fund (hereafter “CAF projects”) in the United States to assess potentially emerging trends in the field of conservation adaptation between 2011 and 2019.”*

Lines 108-115: *“The field of conservation practice is diverse and with varied contours as it is pursued by different organizations, governments, including Indigenous governments, in different regions and contexts globally. While the portfolio of CAF projects*

does not represent the full range of this diversity, it provides a singular dataset of conservation projects that focuses specifically on adaptation objectives and actions across time and as pursued within diverse ecosystems. As such, the portfolio of CAF projects offers a unique “learning laboratory” of adaptation efforts from early adopters and represents an ideal case study to test our typology.”

- 3) We also included a new paragraph in the discussion to address the issue. We discuss the lack of availability to similar datasets, explain that we used the portfolio of CAF projects as proxy to trends in the conservation adaptation, and discuss how further research could apply the R-R-T scale to other contexts.

Lines 438-455: *“Our focus on the portfolio of 104 CAF projects offered a data-rich, unique opportunity to trial a novel framework for conservation adaptation and provide empirical evidence suggestive of trends in conservation practice that have been anecdotally observed globally and across diverse groups^{7-9, 15, 22, 43}. At the same time, these results, building as they do on the activities within one funding portfolio in the United States, come with some important caveats. Foremost is caution in generalizing the findings from this case to the broader field of conservation (something that we do not claim here). To our knowledge, the CAF portfolio is one of the only extensive set of conservation projects that are specifically and intentionally designed to achieve adaptation outcomes. However, the portfolio is embedded within, but is not considered a random subsample of conservation adaptation projects conducted in the United States, nor is it representative of the field of practice. The unavailability of other similar datasets—which speaks directly to the necessity for more empirical assessments of conservation adaptation actions—limited our analysis to the use of the CAF projects as proxy to trends in conservation adaptation funding and implementation over the last decade. Further research is needed to apply the R-R-T scale to explore its application to other adaptation contexts—of particular interest could be its application to a global fund. It could also be used to assess conventional conservation projects with no specific focus on adaptation to elucidate trends in the broader field of conservation, if such a data set could be assembled.”*

3. The authors appropriately claim for

We added details and clarity on our definition of

the ambiguity of the term resilience, but the definitions and examples that they provide could be improved to avoid ambiguity. It seems that from the definitions the authors provide in Figure 1, resistance (active and passive) measures are designated to maintain current or historical status, while resilience actions aim for future recovery. However, it remains unclear the separation of resistance and resilience in point three, given the interdependency of resistance and resilience. For example, actions aimed to re-introduce ecosystems engineers are going to improve the resistance to changes, as well as promote its recovery. Please check Hodgson et al. 2015 TREE, Donohue et al. 2016 EcoLetts, Ingrisich and Bahn 2018 TREE, for more extensive explanations of the interdependency between resistance and recovery. This manuscript would benefit from a clearer distinction between resistance and resilience.	resilience and used the example provided by the reviewer (ecosystem engineers). We also believe that the new Table 1 helps make the distinction between resistance and resilience. Lines 262-275: “The third level, resilience, describes interventions that enhance the capacity of ecosystems to return to desired conditions (past or present) after a disturbance. This definition, which aligns closely with the original ecological meaning of the term³⁰, implies that resilience shares the end goal with resistance of generally limiting changes, but it acknowledges that some changes are unavoidable and sometimes desirable. For instance, restoring forest ecosystems with a diversity of native species can increase resilience (i.e., because there is a greater likelihood that some tree species will continue to persist and function during a disturbance like drought or fire), but it may also lead to changes in community composition. Similarly, the introduction of ecosystem engineers such as beaver can increase the resilience of an ecosystem (i.e., the reconnected floodplain can absorb more water during a flood and reduce flood-related damage to downstream reaches) while also altering historical conditions of streams and valley bottoms^{46, 47}. A survey of published empirical studies⁴⁸ suggests that resilience is commonly used to represent resistance, or recovery, or both. Our application of the term emphasizes recovery, while acknowledging the inevitability of some new elements.”
4. Lines 6-8: This is an incorrect statement. There are plenty of manuscripts exploring the conservation practices applied and their success. One recent example would be Duarte et al. 2020 Nature, but there are many more.	We revised the statement to make it clear that we are referring to conservation adaptation (i.e., focused specifically on adaptation) and not conservation interventions in general in the context of climate change (as reviewed by Duarte et al., 2020 Nature). Lines 7-10: However, little empirical evidence exists on what kinds of conservation actions aimed specifically at climate change adaptation are being implemented in practice, let alone how transformative these actions are.
5. Lines 17-18: Conservation is not only about climate change but global change. The authors should provide here, and throughout the manuscript, a clear statement of the conservation actions targeted in this manuscript.	This is a very helpful and constructive comment. We made sure to clearly identify that we focus on conservation actions aimed at adaptation, and indicated that we would refer to these actions as “conservation adaptation”. Lines 30-37: “Within these and other conservation institutions that operate at multiple scales, novel conservation actions and objectives aimed specifically at helping ecosystems adapt to the accumulating

	impacts of climate change (hereafter “conservation adaptation”) have gained increasing attention⁷⁻⁹. For example, transformative actions such as species translocation and objective setting by triage principles that were not long ago eschewed by most conservation scientists, and declared by practitioners and policy-makers as anathema to the practice of conservation^{10, 11}, are today increasingly highlighted as necessary components of conservation adaptation¹²⁻¹⁶.”
6. Line 27: Could the authors please provide references supporting the increase of the need for building resilience in science and conservation?	We now refer to the three leading conservation organizations in the introduction (IUCN, PEBES and the UN CBD; see answer to comment 1)
7. Lines 31-34: This sentence is very long and difficult to follow, please try to shorten it or split into two.	We split the sentence into two: Lines 46-68: However, few empirical attempts have systematically categorized on-the-ground conservation adaptation projects to assess the extent to which adaptation actions are being implemented, by what approaches, and where they are occurring²⁴. None that we are aware of have assessed how transformative (or not) these actions are in terms of adopting a future-looking approach instead of maintaining current or historical conditions.
8. Line 39: Could the authors be more specific about which trade-offs?	We provided two examples: Lines 70-75: “In addition to providing a much-needed structure and guidance to help practitioners and managers navigate the “sea of adaptation ideas and recommendations”, existing frameworks can also help assess trends in adaptation actions, categorize decisions, and evaluate trade-offs between different strategies (e.g., adopting future-looking approaches vs maintaining current or historical conditions, species vs landscape level conservation).”
9. Line 75: Given the general scope of the journal, it would be worth to define what typologies are. It would help unfamiliar readers to better comprehend the concept and the value of the research done here.	We added a short definition of what we mean by typologies. Lines 150-153: “A precise definition of terms is critical to developing typologies (i.e., classifications based on types of interventions and/or conservation objectives) in rapidly evolving fields of practices, such as contemporary conservation adaptation. These typologies then represent powerful tools that enable comparisons between approaches and outcomes.”

10. Lines 84-102: It would be good to have a more clear description of why the authors discuss these specific typologies.	We added a note that these typologies are arguably the most commonly used and useful. Lines 161-163: The most commonly applied and, arguably, useful typologies based on conceptual hierarchies have categorized conservation adaptation actions on a change continuum ranging from resistance to transformation.
11. Figure 1: The examples posed in this figure would be much more useful if explained in much more detail. Also, for some definitions only one example is provided, giving a narrower view in comparison with other definitions. It would be highly desirable to have a more balanced proportion of examples.	For clarity, we removed the examples from Figure 1. Instead, we included a more detailed table (Table 1; lines 241-243) where we provide two examples of actions and their primary objective for each category of the R-R-T scale (total of 12 examples). We believe that the new structure (i.e., Figure 1 with definitions of the R-R-T categories, Table 1 with detailed examples, and Supplementary Material 2 with examples from the WCS Climate Adaptation Fund) provides more details and clarity.
12. Line 143: It would be beneficial for the manuscript to provide examples of which changes could be desirable.	We added two examples: Lines 264-275: “This definition, which aligns closely with the original ecological meaning of the term³⁰, implies that resilience shares the end goal with resistance of generally limiting changes, but it acknowledges that some changes are unavoidable and sometimes desirable. For instance, restoring forest ecosystems with a diversity of native species can increase resilience (i.e., because there is a greater likelihood that some tree species will continue to persist and function during a disturbance like drought or fire), but it may also lead to changes in community composition. Similarly, the introduction of ecosystem engineers such as beaver can increase the resilience of an ecosystem (i.e., the reconnected floodplain can absorb more water during a flood and reduce flood-related damage to downstream reaches) while also altering historical conditions of streams and valley bottoms^{46, 47}. A survey of published empirical studies⁴⁸ suggests that resilience is commonly used to represent resistance, or recovery, or both. Our application of the term emphasizes recovery, while acknowledging the inevitability of some new elements.”
13. Figure 3: This figure does not provide estimates of the slope of the linear model tendency. At least the authors should provide the slope and if it significantly differs from 0.	We added the trendline equation, and modified the figure for a boxplot to better illustrate the distribution of the RRT scores.

14. Lines 162-164: Please indicate what the numbers in parenthesis represent. Are they mean and standard deviation or are they median and standard error?	We indicated (Lines 318-320; 257) that the numbers in parenthesis are mean (M) and standard deviation (SD).
15. Lines 196-197: This statement is generalising results coming from a very limited sampling size and only one funding body (se general comments).	We made substantial modifications throughout the manuscript to discuss the limitation of our study in generalizing the data (see answers to comment 2). We also deleted the first part of the statement (previously on lines 196-197), and instead provided a short paragraph to summarize the results in the context of the fund: Lines 302-311: “Overall, we find that CAF projects funded prior to 2016 focused primarily on the resistance-resilience end of the R-R-T scale, and those implemented afterwards were more likely to involve transformation. The types of approaches differ across ecosystems with more resistance projects occurring in deserts, grasslands and savannahs, and inland aquatic ecosystems, and more transformative projects in forest, coastal aquatic, and urban/suburban ecosystems. Supplementary Table 1 includes the R-R-T scores for the 104 CAF projects, along with short project descriptions that were crafted, on an annual basis, from full proposals by WCS staff to provide a brief abstract of the projects for the general public. Of the 104 projects funded between 2011 and 2019, the most common categories of actions were resilience (40%), autonomous (26%) and directed (18%) transformation, and active resistance (10%, Figure 2).”
16. Discussion: The discussion is very focused on the results coming from the analyses done from the projects funded by WCS. However, this manuscript would benefit from a more in-depth discussion of the novel R-R-T approach and present the results from the WCS funded projects as a case study.	We made substantial changes to the discussion. In particular, we added two paragraphs (Lines 414-455) discussing the use of the R-R-T scale (see response to comment 2 from Reviewer 3).
17. Line 290: Please cite R software.	We cited R Studio (Line 544).

Reviewer #2 (Remarks to the Author):

This manuscript represents an important contribution to the rapidly evolving field of climate adaptation, particularly in refining a typology for categorizing adaptation approaches on a continuum from resistance (or persistence) to transformation (or change). The manuscript nicely describes the confusion surrounding the overly broad application of the term “resilience”. Of particular interest, however is the development of a single continuous typology that integrates the concepts of passive and active management. I have attempted to do something similar (see Stein et al. 2019 -- DoD adaptation guide) through use of a four quadrat approach (change on one axis and passive/active management along the other), but find the typology presented to be a much more elegant solution and far more receptive to classifying adaptation efforts and charting the shifts in their emphasis over time.

Having done a good job of defining the existing confusion in adaptation typologies and the emphasis on activity rather than outcome- oriented categories, the manuscript tests its application on an empirical data set of adaptation projects. Although this data set has limitations, it is -- to my knowledge -- one of the only extensive sets of projects that are specifically and intentionally designed to achieve adaptation outcomes. For that reason, it is a reasonable proxy in showing trends in adaptation funding and implementation over the decade or so the program has been in existence. The manuscript addresses the one concern that I had in using the data set to chart the shift in the broader community away from resistance and towards more transformational actions -- that is, the evolution over that period of the guidelines and expectations of the funding program and its advisors. That said, the manuscript recognizes that confounding effect, but the analysis confirms that there has indeed been a resulting shift toward more innovative and transformation-oriented projects, bolstering the conclusion that a shift in perception -- if not in the broader community then at least in this important part of the natural resource-oriented adaptation community -- is underway.

Comments	Answer
1. The one thing that I was expecting to see, but is missing, was any reference to a similar/related effort underway in the community towards development of the so-called “RAD” (Resist, Accept, Direct) framework, which significantly overlaps with RRT scale described here. It may be that there are no citable accounts yet for RAD, but at a minimum I would expect some recognition of that emerging framework and discussion of how RRT is similar to or different than RAD.	This is an excellent point. We were aware of the RAD framework, but there are still not peer-reviewed publications that we could cite in this manuscript. In order to stay as much as possible within the word limit, we added a short description of the RAD framework and the main difference with the RRT. We cited the first paper recently published (2020) on the RAD framework. Line 163-183: “This continuum typically involves three categories: (1) resisting changes in order to maintain current conditions (Resistance); (2) improving the capacity of a system to return to desired conditions after disturbance (Resilience); (3) allowing and/or facilitating the transition to new conditions (Transformation)^{18, 33, 36-38}. Other similar terminologies also exist—for instance, North American researchers and agencies have proposed the Resist, Accept, or Direct (RAD) framework³⁹. The main difference between the two typologies is the intermediate category: Resilience focuses on enhancing the capacity of a system to return to a

desired state, whereas Accept refers to allowing changes to occur without interference. The RAD framework glosses over the less-transformative end of the spectrum, by leaving resilience out entirely and not differentiating between resistance and resilience.”

We also refer to the “accept” category later in the manuscript when describing the fourth level of our scale (autonomous resistance):

Line 289-291: *The last three levels represent different degrees of transformation. Autonomous transformation describes actions aimed at allowing for changes without actively shaping the projected transformation³³, which is equivalent to the RAD framework’s Accept category³⁸.*

I strongly support publication and view this as an important contribution to the evolving science and practice of climate adaptation.

Reviewer #3 (Remarks to the Author):

This manuscript contributes evidence of evolution within the field of climate adaptation. I have a few concerns that could be addressed with revision, but generally feel the manuscript worthy of publication. I find the development of the typology to be an improvement of more coarse Resistance-Resilience-Transition comparisons that I have seen published, and will have broad utility. Further, the application of the typology to a subset of projects is informative of reasonably real trends in both funding and project development directions, even if only a segment of a much larger population of conservation projects. It is in my opinion that the manuscript be considered for publication following revision to address the following concerns.

Comments	Answer
1. My major concern is whether the trends documented in this paper are observed in the broader conservation world. As I read the last paragraph of the introduction (and being unfamiliar with the project proposal and selection process for the Wildlife Conservation Society’s Climate Adaptation Fund), I wondered whether the project selection process increasingly emphasized transformative projects over time or prioritized those types of projects? After reading the discussion, I realize the role of funding limitations is an important factor. However, my instinct is that the trend is real. Is there additional evidence that the authors could point to that indicate that the trends here may very well be consistent with trends in the broader conservation realm? Within the ‘restoration’ program I work with, I have noticed a transition in the types of projects considered for funding. More specifically, there seems to be a wider range of the types of projects proposed in more recent years than historically. This may partially be an effect of enhanced understanding of the ecosystem structure and function, demographic changes within the scientist and manager community that participate in the program, or the increased awareness of resilience and adaptation ideas. Anecdotally, I do think the application of the typology would see an increase from resistance to resilience in the ecosystem the program focuses on.	This is a very valid point. We also observe similar anecdotal evidence that this shift is happening in our own practice. We attempted at many places in the manuscript to acknowledge the fact that this study is not meant to be representative of the broader field of conservation (see our answer to comment 2 from Reviewer 1). We also added references from the literature where they discuss a potential shift towards transformation, for instance: Line 378-381: “Reflecting broader conversations about the need for more transformation actions in conservation adaptation^{7, 8, 22}, earlier CAF projects (prior to 2016) were more likely to include approaches designed to resist changes, whereas more recent projects tended towards managing for transformative change.” Line 438-441: “Our focus on the portfolio of 104 CAF projects offered a data-rich, unique opportunity to trial a novel framework for conservation adaptation and provide empirical evidence suggestive of trends in conservation practice that have been anecdotally observed globally and across diverse groups^{7-9, 15, 22, 43}.” We also acknowledge that finding additional datasets on which the framework could be tested is a priority next step (albeit beyond the scope of this study): Line 451-455: Further research is needed to apply the R-R-T scale to explore its application to other adaptation contexts—of particular interest could be its application to a global fund. It could also be used to assess conventional conservation projects with no specific focus on adaptation to elucidate trends in the broader field of conservation, if such a dataset could be assembled.

2. I do think the typology can be useful in understanding the trends and directions within conservation practices; however, I think there is still ambiguity in the application of the typology that could be clarified. For example, I work in riverine ecosystems and using the typology developed, have a difficult time selecting a single term for some of the management actions that are often considered. Enhancing longitudinal (upstream/downstream) connectivity could be considered resistance in that it is an action designed to reestablish historical connectivity, or it could be considered resilience as it improves the capacity of the system to reorganize following a disturbance, or it could be viewed as transformative to facilitate a transition to a new structure and function through enabling species movements. Is the typology applied to the primary objective of the project even if it may ultimately have a different effect? Further clarification in the methods regarding how the typology coding was applied could be beneficial.

This is a very useful comment that is in line with a few comments from Reviewer 1. We attempted to clarify the use of our scale in many places in the manuscript.

First, we provided more details on our definition of resilience:

Line 262-275: “The third level, resilience, describes interventions that enhance the capacity of ecosystems to return to desired conditions (past or present) after a disturbance. This definition, which aligns closely with the original ecological meaning of the term³⁰, implies that resilience shares the end goal with resistance of generally limiting changes, but it acknowledges that some changes are unavoidable and sometimes desirable. For instance, restoring forest ecosystems with a diversity of native species can increase resilience (i.e., because there is a greater likelihood that some tree species will continue to persist and function during a disturbance like drought or fire), but it may also lead to changes in community composition. Similarly, the introduction of ecosystem engineers such as beaver can increase the resilience of an ecosystem (i.e., the reconnected floodplain can absorb more water during a flood and reduce flood-related damage to downstream reaches) while also altering historical conditions of streams and valley bottoms^{46, 47}. A survey of published empirical studies⁴⁸ suggests that resilience is commonly used to represent resistance, or recovery, or both. Our application of the term emphasizes recovery, while acknowledging the inevitability of some new elements.”

Second, we added one paragraph to discuss the issues raised by the reviewer regarding the possibility that a project/action could be classified in more than one category, and explained how we dealt with the issue (i.e., by coding the primary objective of the projects).

Line 414-427: “Although six mutually-exclusive categories comprise the linear R-R-T scale, we acknowledge that the distinction between categories (e.g., passive resistance, resilience and autonomous transformation) is not always black and white. In particular, the potential outcomes of adaptation actions may overlap two or more categories. For instance, a project that enhances longitudinal

connectivity of forested or aquatic systems may ultimately result in re-establishing historical connectivity⁵⁵ (active resistance), enhancing the capacity of a system to respond to disturbance (resilience) and/or facilitating the migration of species to new areas⁵⁶ (passive transformation). Similarly, the reintroduction of ecosystem engineers (e.g., beavers) may allow a system to better resist changes, but it may also enhance its resilience by promoting recovery after natural disturbances^{47, 57}. To prevent this ambiguity, we classified projects based on the primary objective identified in the proposals with regards to desired future conditions (and how similar or different they are from current or historic conditions). The R-R-T scale could also be used to assess the on-the-ground outcomes of adaptation actions after their implementation, which could lead to situations where one project could be classified into multiple categories at once.

Third, we added a paragraph in the method section to indicate that we coded the projects based on their primary objective, and referred to the details provided in the discussion (as outlined above):

Line 507-512: *“We carried out a content analysis⁶⁶ of full proposals for projects that were awarded grants from the Climate Adaptation Fund (CAF) to categorize the portfolio of CAF projects using the qualitative analysis software NVIVO (version 12.6.0) and coding project documents directly using the R-R-T scale. To ensure consistency, reduce ambiguity and capture the motivation behind adaptation actions, each project was assigned one of the six categories of the R-R-T scale based on their primary objective.”*

REVIEWERS' COMMENTS:

Reviewer #2 (Remarks to the Author):

In reviewing the revised manuscript, I found that the authors appropriately and adequately addressed my comments on the original submission, as well as the comments provided by other reviewers.

Regarding the inclusion of the RAD framework as a related typology, a citable publication has appeared between the time of original submission and submission of the revised manuscript and the authors describe that framework and contrast it with their RRT typology. One note I would make is that the RAD framework emphasizes manager "actions" (i.e., verbs) while the RRT framework is a combination of actions and outcomes.

Other reviewers questioned the reliance on the WCS Adaptation Fund project portfolio, but I would emphasize what the authors have described, which is that this is a unique data set. That said, the revision appropriately notes that this is intended as a case study, rather than as a sample of the field as a whole.

I support publication of this revised manuscript.

Reviewer #3 (Remarks to the Author):

I reviewed the responses to reviewer comments and am highly satisfied with the changes to the manuscript. The authors thoughtfully responded to reviewer comments by incorporating those ideas and text into the manuscript. I highly recommend for publication.

Responses to Reviewers

Dear Reviewers,

Thank you for your positive review on our manuscript now titled: “R-R-T (Resistance-Resilience-Transformation) typology reveals differential conservation approaches across ecosystems and time”. We have gone through your reports and have outlined below our response to your comment, with additions to the manuscript highlighted in grey.

Best regards.

Reviewer #2:

In reviewing the revised manuscript, I found that the authors appropriately and adequately addressed my comments on the original submission, as well as the comments provided by other reviewers.

Comments	Answer
Regarding the inclusion of the RAD framework as a related typology, a citable publication has appeared between the time of original submission and submission of the revised manuscript and the authors describe that framework and contrast it with their RRT typology. One note I would make is that the RAD framework emphasizes manager "actions" (i.e., verbs) while the RRT framework is a combination of actions and outcomes.	As suggested, we modified the text to indicate the main difference between the R-R-T scale and the RAD framework. Lines 123-130: The main difference between the two typologies is that the RAD framework emphasizes on managers' actions, whereas the R-R-T framework proposes a combination of actions and outcomes. Also, there is a difference between the intermediate category: Resilience (R-R-T framework) focuses on enhancing the capacity of a system to return to a desired state, whereas Accept (RAD framework) refers to allowing changes to occur without interference. The RAD framework glosses over the less-transformative end of the spectrum, by leaving resilience out entirely and not differentiating between resistance and resilience.

Other reviewers questioned the reliance on the WCS Adaptation Fund project portfolio, but I would emphasize what the authors have described, which is that this is a unique data set. That said, the revision appropriately notes that this is intended as a case study, rather than as a sample of the field as a whole.

I support publication of this revised manuscript.

Reviewer #3

I reviewed the responses to reviewer comments and am highly satisfied with the changes to the manuscript. The authors thoughtfully responded to reviewer comments by incorporating those ideas and text into the manuscript. I highly recommend for publication.